# Urinary Dipstick Is Not Reliable as a Screening Tool for Albuminuria in the Emergency Department—A Prospective Cohort Study

**DOI:** 10.3390/diagnostics12020457

**Published:** 2022-02-10

**Authors:** Christian B. Nielsen, Henrik Birn, Frans Brandt, Jan D. Kampmann

**Affiliations:** 1Department of Internal Medicine, University Hospital of Southern Denmark, Sydvang 1, 6400 Sonderborg, Denmark; frans.brandt.kristensen@rsyd.dk (F.B.); jdk@rsyd.dk (J.D.K.); 2Department of Clinical Medicine, Aarhus University, Palle Juul-Jensens Boulevard 82, 8200 Aarhus, Denmark; hb@biomed.au.dk; 3Department of Biomedicine, Aarhus University, Høegh-Guldbergs Gade 10, 8000 Aarhus, Denmark; 4Department of Renal Medicine, Aarhus University Hospital, Palle Juul-Jensens Boulevard 99, 8200 Aarhus, Denmark; 5Institute of Regional Health Research, University of Southern Denmark, Campusvej 55, 5230 Odense, Denmark

**Keywords:** urinary dipstick, proteinuria, albuminuria, screening, kidney disease, emergency department

## Abstract

Albuminuria is a sensitive marker for renal dysfunction. Urinary dipstick tests are frequently used to screen for urinary abnormalities in the emergency department (ED). The aim of this prospective cohort study is to evaluate the usefulness of urinary dipstick testing as a screening tool for albuminuria in the ED setting and to determine the persistency of albuminuria identified in the acute setting. Urinary dipstick tests and spot urine samples were obtained simultaneously for analysis of the urinary albumin-creatinine ratio (ACR). Participants with positive dipsticks for protein were invited for a second urinalysis four to six weeks after admission. The study included 234 patients admitted to the ED. Urinalysis was performed on 178 patients of which 46% (n = 82) had positive urinary dipstick tests for proteinuria. The sensitivity and specificity of the dipstick test were low (72.7% and 55.7% respectively) when compared to the ACR. Of the 82 patients with positive dipsticks at admission, 35 were available for follow-up. We observed a significant reduction in ACR at follow-up when compared to ACR at admission (*p* = 0.004). This paper concludes that urinary dipstick tests are not a reliable means to screen for albuminuria in the ED setting.

## 1. Introduction

Urinary dipstick tests are low-cost diagnostic tools which provide point-of-care semi-quantitative information about protein, ketones, glucose, leukocytes, the presence of nitrite, and urinary pH [1]. In the emergency department (ED) urinary dipsticks are often used as simple tools to identify urinary tract infections [2]. 

Proteinuria refers to increased excretion of all proteins through the urine, while albuminuria refers to increased urinary excretion of albumin, which is the predominant urinary protein [3]. Albuminuria is a sensitive marker for renal dysfunction and has diagnostic, therapeutic, and prognostic significance [4,5]. There is a linear correlation between albuminuria and all-cause mortality, as well as cardiovascular mortality independent of estimated GFR (eGFR) [6]. Screening for albuminuria may be relevant since antiproteinuric drugs such as angiotensin receptor blocker (ARB) and angiotensin-converting-enzyme inhibitor (ACEI) can slow down the progression of both diabetic and nondiabetic kidney disease by inhibiting the renin-angiotensin-aldosterone-system (RAAS) [7]. Whether or not a urinary dipstick test may be useful as a screening tool for persistent albuminuria in the ED depends on the following two factors: firstly, the dipstick’s ability to detect albuminuria; and secondly, the appropriateness of the ED as a setting for performing the screening. 

Current guidelines on chronic kidney diseases (CKD) from the Kidney Disease Improving Global Outcomes (KDIGO) recommend measuring albuminuria to quantify proteinuria since albumin is the predominant protein in the vast majority of proteinuric kidney diseases [8]. A timed urine sample allowing estimation of the albumin excretion rate (AER) is the gold standard for quantifying albuminuria. However, in many circumstances the collection of urine for AER is impractical. Studies have shown a strong correlation between AER and the albumin-creatinine ratio (ACR) and thus ACR is considered reliable for quantifying albuminuria in most cases [9,10,11]. The American Diabetes Association recommends that a semi-quantitative or qualitative screening to detect albuminuria should have a sensitivity above 95% [3,12]. This guideline is based on diabetic patients, although we suggest that this cut-off value for a valid screening tool is fair to use in non-diabetic patients as well. 

Multiple non-renal conditions are associated with a temporary increase in urinary excretion of albumin including fever, stress, lower urinary tract infection, and exercise [3,13,14]. Such conditions are common in the ED. In addition, other factors such as a concentrated urine specimen, hematuria, or the presence of iodinated radio-contrast, mucus, leukocytes, semen, or vaginal secretion can cause a positive result on the dipstick without representing renal dysfunction [13,15]. 

To our knowledge, no studies have compared urinary dipstick tests for protein with ACR in the ED setting, nor have any studies compared individual dipstick tests with ACR results after discharge from the ED. Thus, the aim of this study is to evaluate the usefulness of using urinary dipstick tests as a screening tool for albuminuria in the ED setting and to determine to what extent albuminuria identified in the ED is persistent.

## 2. Materials and Methods

### 2.1. Study Design and Population

This was a prospective cohort study that included all patients above the age of 18 admitted to the ED under internal medicine or cardiology at the Hospital of Southern Denmark, Sønderborg. Patients were included from 27 March 2019 to 23 June 2019. Prior to consent, patients were informed both orally and in written form, preferably within 24 h of arrival to the ED. 

Patients with a positive dipstick test were asked to return for a second urinalysis four to six weeks after admission unless they had been hospitalized within three weeks prior to the follow-up analysis. 

During admission, a single mid-stream urinary sample was obtained from all patients both for dipstick and ACR analyses. A urine specimen obtained between 4:00 a.m. and 10:00 a.m. was defined as morning urine. Urine specimen collection, dipstick analysis, and transfer of the spot urine sample for albumin concentration (U-albumin) and ACR were managed by nurses in the ED following written instructions. These included cleaning the genital area prior to urination. The follow-up analyses were performed at the Department of Clinical Chemistry in Sønderborg following the same instructions as those given to the nurses in the ED. For urinary dipstick analyses we used a Siemens Multistix 7 analyzed by Siemens Clinitek Status, which reports semi-quantitative values for blood, ketones, glucose, protein and leukocytes, as well as positivity for nitrite and a urinary pH measurement. The dipstick protein result is based on a colorimetric reaction [1]. It predominantly detects albumin as it contains most of the amino groups and, to a lesser extent, globulins [8]. The dipstick is reported to be sensitive for U-albumin concentrations >150 mg/L [1]. We used +1 for protein as the cut-off for a positive dipstick result, which corresponds to a urinary albumin concentration of >300 mg/L [4]. 

U-albumin and U-creatinine were analyzed using a standardized, automatized routine biochemical assay on a Roche Cobas 702 at the local Department of Clinical Chemistry. The ACR was calculated by dividing the urinary albumin concentration (mg/L) by the urinary creatinine concentration (g/L). U-albumin was reported as mg/L and ACR was reported as mg/g. According to KDIGO guidelines, albuminuria is categorized as follows: normal to mild albuminuria (ACR < 30 mg/g); moderate albuminuria (ACR 30–300 mg/g); and severe albuminuria (ACR > 300 mg/g).

### 2.2. Clinical and Demographic Characteristics

All clinical and demographic data were obtained from patient records and recorded into an encrypted Microsoft Access database along with results of the urinary analyses. Prior to urinalysis we collected information about sex, age, smoking habits and diagnosis on admission; the most recently recorded body temperature, blood pressure (BP) and plasma-creatinine; estimated glomerular filtration rate (eGFR) calculated using the CKD-EPI formula without correction for race [16]; and plasma-C-reactive protein (P-CRP) and white blood cell count (WBC). In addition, we collected information concerning any previous proteinuria (defined as a previously recorded positive dipstick); previous diagnoses of diabetes mellitus (DM) or CKD (defined as the patients in which CKD had previously been recorded as a diagnosis); and heart disease (including heart failure, ischemic heart disease, arrhythmia, and heart valve diseases), hypertension and/or the use of either ACEIs, ARBs, prednisolone or non-steroid anti-inflammatory drugs (NSAID). Fever was defined as a body temperature above 38 °C and high BP as a systolic BP above 140 mmHg. 

### 2.3. Statistical Analysis

We calculated median values for age, P-creatinine, P-CRP, WBC, eGFR, ACR, and U-albumin with their respective interquartile ranges (IQR) together with percentage estimates of patients with hypertension, fever, known heart disease, DM or CKD; previous proteinuria, smoking habits, male sex; and admission under internal medicine as well as any current use of ACEI, ARB, prednisolone, and NSAIDs. Patients were divided into four subgroups according to their dipstick readings, designated as either 0 and/or trace (non-proteinuric), 1+, 2+, or 3+. The *p*-values for percentage estimates were calculated using a Chi-square test comparing proteinuric sub-groups (1+ and 2+) with the non-proteinuric sub-group (0 or trace). As a result of the small sample size, the 3+ proteinuric sub-group did not fulfill Cochran’s rule and thus, Fischer’s test was applied instead of the Chi-square test. The remaining results were reported as median values with their respective IQRs. Proteinuric groups (1+, 2+, and 3+) were compared with the non-proteinuric group using a *t*-test. Since eGFR was not normally distributed, this was compared using the Wilcoxon rank sum test.

A Z-test was used in order to calculate the odds ratios (OR) for having severe albuminuria with a positive dipstick protein result, respectively 1+, 2+, and 3+, compared with a negative dipstick result (0 and trace). 

Sensitivity, specificity, positive predictive values (PPV), and negative predictive values (NPV) for dipstick results were calculated using two different cut-off values of ≥1+ and ≥3+ for protein respectively, with ACR ≥ 300 mg/g as the standard reference representing severe albuminuria. In addition, a receiver operating characteristic curve (ROC-curve) was calculated using dipstick results 1+, 2+, and 3+ for protein as cut-off values and ACR > 300 mg/g as the standard reference.

A Wilcoxon signed-rank test was performed in order to compare ACR in the follow-up group with ACR from initial admission.

## 3. Results

A total of 243 patients were included in the study. Of these, 65 patients missed the urinalysis before transfer to other wards or discharge and were therefore excluded from analyses (Figure 1). Of the 178 patients included in the analyses, 82 (46%) had positive dipsticks for protein, of which 9.8% had severe albuminuria (ACR > 300 mg/g). The urine specimens from 60 patients were morning urine (obtained from 4:00 a.m. to 10:00 a.m.); these returned positive dipsticks for protein in 18.2% (n = 11) of the cases. 

The median CRP levels were significantly higher in all the proteinuric sub-groups (1+, 2+, and 3+; *p* < 0.05, *t*-test) when compared with the non-proteinuric group. The median e-GFR levels were significantly lower in the 3+ sub-group (3+; *p* = 0.03, *t*-test) but not significant in the 1+ and 2+ sub-groups (1+; *p* = 0.38, 2+; *p* = 0.09 *t*-test), in comparison with the non-proteinuric group. There were no significant differences in WBC comparing the proteinuric sub-groups with the non-proteinuric sub-group (1+; *p* = 0.13, 2+; *p* = 0.20, 3+; *p* = 0.31, *t*-test) (Table 1). The mean ACR and U-albumin concentrations in the entire study population were 105.0 mg/g (95%CI: 68.1–141.9 mg/g) and 86.2 mg/L (95%CI: 51.6–120.8 mg/L), respectively. Both ACR and U-albumin concentrations increased across the sub-groups (Table 1) and the differences were significant in all the proteinuric sub-groups (1+, 2+, and 3+; *p* < 0.05, *t*-test) when compared with the non-proteinuric sub-group. We cross-tabled the ACR results with the results of the dipstick for protein using the sub-categories normal to mild albuminuria (ACR > 30 mg/g), moderate albuminuria (ACR 30–300 mg/g), severe albuminuria (ACR > 300 mg/g) and for the dipstick sub-groups 0 to trace, 1+, 2+, and 3+, both at admission (Table 2) and at follow-up (Table 3). 

Eleven patients had severe albuminuria (ACR > 300 mg/g) at admission. Of these patients, five had 3+, two had 2+, and one had 1+ for protein. The last three patients had negative dipsticks for protein despite having severe albuminuria (ACR 538 mg/g, 339 mg/g, and 1117 mg/g) and their U-albumin concentrations were 44 mg/L, 71 mg/L, and 455 mg/L, respectively. One of these patients had known CKD. Of the five patients with dipstick 3+ for protein, four patients also had 3+ for blood. Twenty-eight percent (n = 23) had an ACR < 30 mg/g despite a dipstick showing >1+ (n = 82), while 28.1% (n = 27) with an ACR > 30 mg/g had a negative dipstick (Table 2).

The OR for severe albuminuria (ACR > 300 mg/g) provided by a 3+ on the dipstick for protein was 155 when compared to a negative dipstick, which was significant (*p* < 0.05, Z-test). The ORs for severe albuminuria provided by a urinary dipstick of 1+ or 2+ for protein compared with a negative dipstick were 0.61 and 2.82, respectively, which were not significant (1+; *p* = 0.67, 2+; *p* = 0.272, Z-test) (Table 4). 

### 3.1. Sensitivity, Specificity, and Positive and Negative Predictive Values

We estimated the accuracy of the dipstick test using ≥1+ as the cut-off to identify an ACR > 300 mg/g showing a sensitivity of 72.7% and a specificity of 55.7%. The PPV was 9.8% and the NPV was 96.9%. When using ≥3+ as the cut-off value, the sensitivity was 45.5% and the specificity was 99.4%. The PPV was 83.3% and the NPV was 96.5%. The area under the ROC curve was 0.76 with 1+ being closest to the ideal test point (Figure 2).

### 3.2. Follow-Up

Among the 82 patients scheduled for follow-up, four died, nine patients were recently (within three weeks prior to the analysis) or currently admitted to the hospital, while 34 patients did not show up. Consequently, 35 patients were available for follow-up analyses (Figure 1). Of these, 19 had positive dipsticks at follow-up (Table 3). Eight of those patients had not previously been diagnosed with CKD and none had severe albuminuria (ACR > 300 mg/g). One patient had moderate albuminuria (ACR 125 mg/g) despite a negative dipstick and had no known history of CKD. Of the 18 patients with elevated ACR at admission, 10 had also an elevated ACR at follow-up; meanwhile, no patients with a normal ACR at admission had an elevated ACR at follow-up (Table 5).

Overall, a reduction in ACR at follow-up was identified in 68.6 % (n = 24) of patients and ACR was significantly lower at follow-up when compared to at admission (*p* = 0.004, Wilcoxon).

## 4. Discussion

Our results suggest that urinary dipstick evaluation is not an appropriate tool to identify albuminuria as defined by ACR in the ED based on low sensitivity and low specificity. Notably, the low sensitivity suggests that a negative dipstick for protein does not exclude severe albuminuria. 

The odds ratio showed a higher risk of having severe albuminuria with a negative dipstick for protein than if the dipstick was 1+. Although this was not significant, it is consistent with the inaccuracy of dipstick results in the range of 1+ for protein and negative. Using >3+ as the cut-off value for detecting severe albuminuria further reduced sensitivity and increased specificity. A dipstick 3+ for protein was associated with a high odds ratio for severe albuminuria when compared with a negative dipstick. Thus, a finding of 3+ on a dipstick in patients admitted to the ED with no previous history of severe albuminuria should lead to further investigations, such as ACR. This may still be relevant in the ED where the urinary dipstick test is still frequently used, mostly in relation to suspected UTI [17]. This suggestion, however, is based on the small number of patients (n = 6) with 3+ in this study.

We observed a lower incidence of proteinuria from dipstick tests comparing morning urine with urine from the rest of the day. We are unable to tell whether this was due to a coincidence or maybe as a result of less excretion of protein during the night/morning. However, this is interesting since KDIGO recommends using morning spot urine samples as screening for albuminuria [7]. 

Few studies have compared the sensitivity and specificity of urinary dipstick tests and ACR for the detection of urinary protein [18,19,20,21]. A previous study in Korean adults (n = 20,759) also identified a low sensitivity of 68.2% when using a urinary dipstick >1+ to detect severe albuminuria (ACR > 300 mg/g) in patients aged ≥60 years [19]. However, the researchers identified a high specificity of 99.4%. Interestingly, the sensitivity remarkably increased to 96.9% in the age group 20–39 years. The authors have not explained a possible cause for this difference. Another study involving Japanese adults (n = 2321) with a mean age of 64 years identified a similar sensitivity of 76.9% and a specificity 96.5% [21]. A study of Australian adults (n = 11,247) with a mean age of 51.6 years showed a high sensitivity of 98.9% and a specificity of 92.6%. This sensitivity was similar in both the <50 and >50 age groups [20]. The major difference between our study and the aforementioned studies is in cohort selection. The mentioned studies are based on general populations, whereas our study is based in the ED setting. Surprisingly, we still identified similar sensitivity with the Japanese and Korean cohorts, but not the high sensitivity identified with the Australian cohort. The low sensitivity in our study was caused by three patients having severe albuminuria, despite negative dipsticks for protein. In two of these cases, we identified low U-albumin concentrations and correspondingly low U-creatinine, suggesting diluted urine. However, information on the concentration levels of urine among similar populations in the ED setting is lacking and we are, therefore, unable to tell whether diluted urine may be more present in the ED setting compared to the general population. 

The prevalence of dipsticks >1+ for protein in the Japanese, Korean, and Australian studies were 1.2%, 2.6%, and 8.1%, respectively. These differences may be explained by the use of different types of equipment used for analyses of dipsticks. Interestingly, the prevalence of dipsticks >1+ in this study was 46.1%. All the above-mentioned studies showed specificities close to 100%. The specificity of this study was 55.7 %, which may be related to the high prevalence of dipsticks >1+ for protein resulting in many false positives. Sensitivity and specificity are reciprocal to each other [22]. Based on our low specificity compared to those in the above-mentioned studies, one would expect our sensitivity to be significantly higher. However, that was not the case. As mentioned earlier, factors such as acute illness, fever, and stress may increase albumin excretion, resulting in a higher prevalence of dipsticks >1+ compared to the general population [13]. However, this should not affect specificity and sensitivity. Specificity can be affected by factors such as hematuria and leukocytes in the urine, as seen in UTIs [13]. Oliguria, second to dehydration, may result in increased albumin concentration in the urine, thus leading to a positive dipstick despite a normal ACR [23]. This may be more prevalent in the ED compared to the ambulatory setting. Taking into account that UTIs are more prevalent in the ED in comparison with the general population, a lower specificity of dipstick tests is expected in the ED setting, as seen in this study. Unfortunately, as a result of the small sample pool at follow-up (n = 35), sensitivity and specificity were not determined at follow-up for comparison.

Our study showed a significant reduction in albuminuria among patients at follow-up. Almost half of the patients with an elevated ACR (>30 mg/g) had a normal ACR at follow-up (ACR < 30 mg/g). No patients with a normal ACR in the ED displayed an elevated ACR at follow-up. Furthermore, we demonstrated a significant correlation between higher CRP levels and albuminuria, supporting the suggestion that acute inflammation may increase albumin excretion. This may explain why transient albuminuria was frequently observed in this study. 

The high prevalence of positive dipstick results together with low sensitivities and specificities would make screening for albuminuria very time-consuming and cost-ineffective.

### 4.1. Strengths

The strengths of this study include the use of a prospective cohort study with detailed baseline information. We included an unselected patient population that we believe mirrors the average patient admitted to the ED. We used standard operating procedures for both urinalysis and urine collection. 

By having a follow-up group, we were able to demonstrate the correlation between acute illness and increased excretion of albumin in the urine. 

### 4.2. Limitations

The study took place at a single center with a small sample size for urinalysis (n = 178). Only 11 patients had severe albuminuria, limiting the accuracy of the sensitivity assessment. Of the follow-up group (n = 82), less than half actually attended follow-up (n = 35), limiting the validity of the follow-up assessment. Furthermore, patients with recent hospitalization or death were excluded from the follow-up. This may represent selection bias, as one could expect these patients to be more likely to still have albuminuria. 

Patients invited to the follow-up were limited to those who had positive dipstick tests. Patients with negative dipsticks and actual albuminuria (ACR > 30 mg/g) were not included in the follow-up. Additional studies are required to validate findings in a larger cohort. 

## 5. Conclusions

The urinary dipstick test for proteinuria in the ED showed both low sensitivity and specificity. We displayed a significant association between high CRP levels, fever, and a positive dipstick, suggesting that albuminuria in the ED is often transient. This makes the ED an inappropriate setting to screen for persistent albuminuria.

## Figures and Tables

**Figure 1 diagnostics-12-00457-f001:**
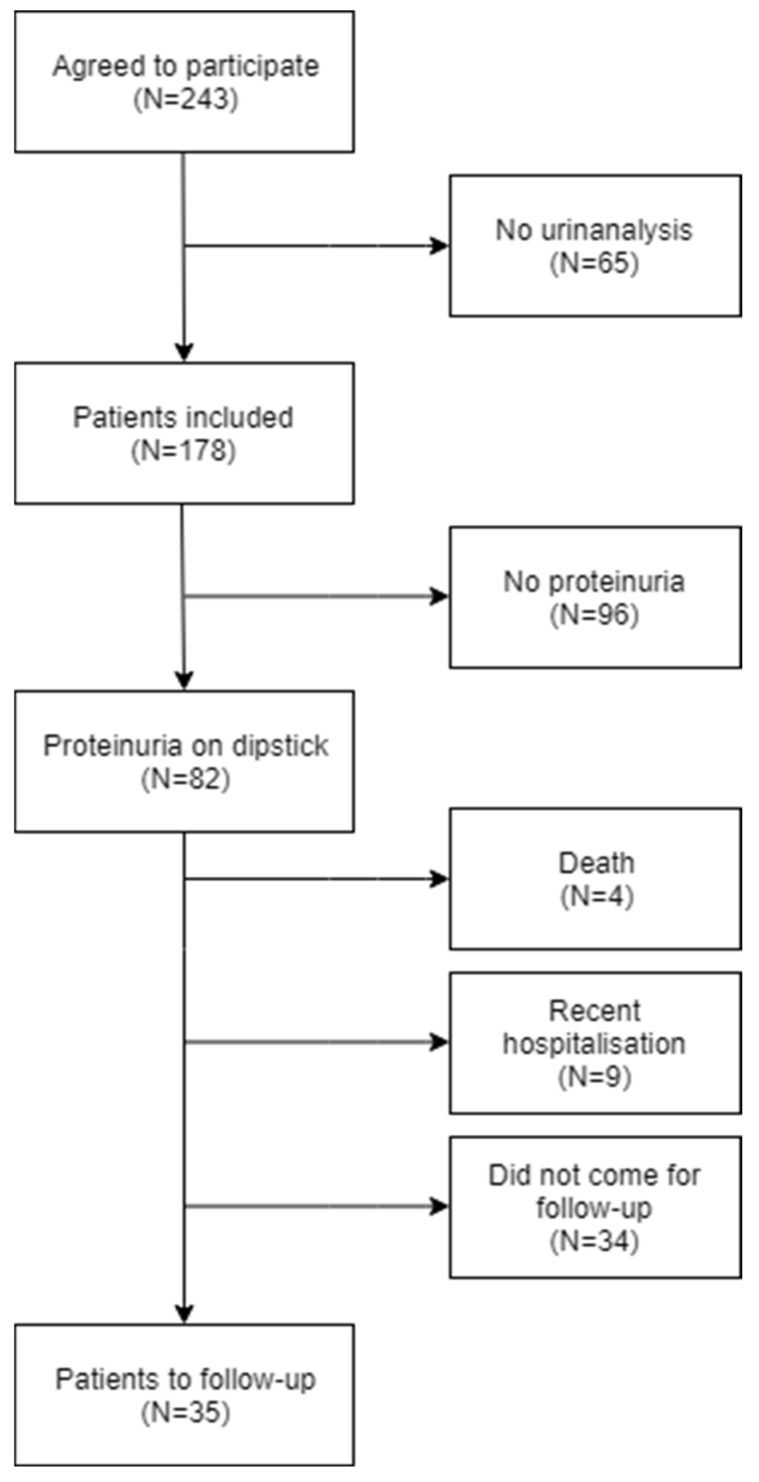
Flow chart of the selection of patients. Prevalence of the comorbidities mentioned above increased across the sub-groups with increasing dipstick proteinuria (Table 1).

**Figure 2 diagnostics-12-00457-f002:**
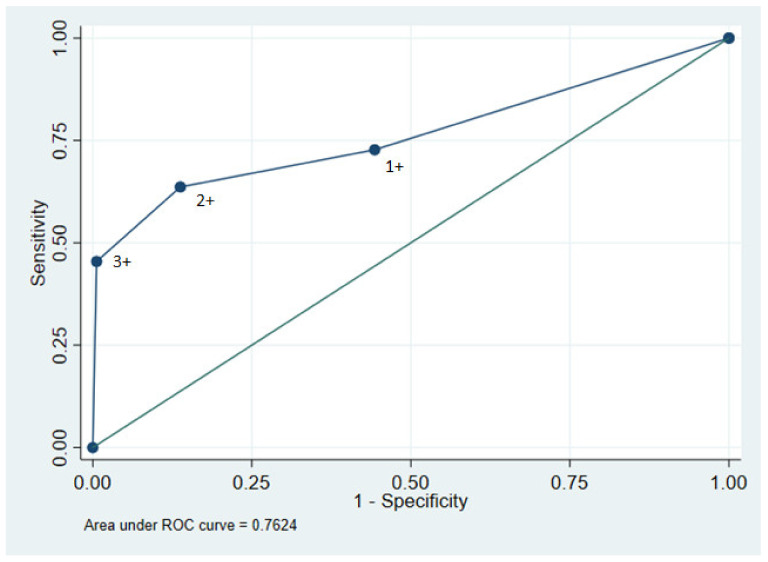
ROC curve for urinary dipstick using 1+, 2+, and 3+ as cut-off values and ACR > 300 mg/mL as reference value. Area under the curve yielded 0.7624.

**Table 1 diagnostics-12-00457-t001:** Demographic and clinical characteristics of the study population (n = 178) were divided according to dipstick results.

Dipstick					
	Sub-Groups				Total
	0 or +/−	1+	2+	3+	
Patients (n)	96	52	24	6	178
Men, %	54.2 (n = 52)	75.0 (n = 39), *p* = 0.013	70.8 (n = 17), *p* = 0.14	83.3 (n = 5), *p* = 0.22	63.5 (n = 113)
Age	71 (59.5, 78.5)	71.5 (64.5, 79.5) *p* = 0.49	74 (58.5, 78.5) *p* = 0.67	78 (73, 81) *p* = 0.065	72 (61.3, 79.0)
Internal medicine pt., % ^1^	65.6 (n = 63)	90.4 (n = 47), *p* < 0.001	91.7 (n = 22), *p* = 0.012	66.7 (n = 4), *p* = 1.00	76.4 (n = 136)
Fever, % ^2^	5.2 (n = 5)	19.2 (n = 10), *p* = 0.007	12.5 (n = 3), *p* = 0.20	0.0 (n = 0), *p* = 1.00	10.1 (n = 18)
Hypertension, %	58.3 (n = 56)	63.5 (n = 33), *p* = 0.54	79.2 (n = 19), *p* = 0.059	100.0 (n = 6), *p* = 0.079	64.0 (n = 114)
Diabetes, %	19.8 (n = 19)	23.1 (n = 12), *p* = 0.64	25.0 (n = 6), *p* = 0.57	33.3 (n = 2), *p* = 0.60	21.9 (n = 39)
Heart disease, %	43.8 (n = 42)	50.0 (n = 26), *p* = 0.47	50.0 (n = 12), *p* = 0.58	83.3 (n = 5), *p* = 0.092	47.8 (n = 85)
CKD, %	5.2 (n = 5)	15.4 (n = 8), *p* = 0.037	37.5 (n = 9), *p* < 0.001	66.7 (n = 4), *p* < 0.001	14.6 (n = 26)
Previously proteinuria, %	28.1 (n = 27)	28.8 (n = 15), *p* = 0.93	45.8 (n = 11), *p* = 0.095	83.3 (n = 5), *p* = 0.011	32.6 (n = 58)
Smoker, %	20.8 (n = 20)	19.2 (n = 10)	29.2 (n = 7)	16.7 (n = 1)	21.3 (n = 38)
ACEI, %	17.8 (n = 17)	21.2 (n = 11), *p* = 0.61	41.7 (n = 10), *p* = 0.012	16.7 (n = 1), *p* = 1.00	21.9 (n = 39)
ARB, %	29.2 (n = 28)	23.1 (n = 12), *p* = 0.43	29.2 (n = 7), *p* = 1.00	33.3 (n = 2), *p* = 1.00	27.5 (n = 49)
Prednisolone, %	8.3 (n = 8)	9.6 (n = 5), *p* = 0.79	8.3 (n = 2), *p* = 1.00	16.7 (n = 1), *p* = 0.43	9.0 (n = 16)
NSAID, %	10.4 (n = 10)	5.8 (n = 3), *p* = 0.34	12.5 (n = 3), *p* = 0.77	0.0 (n = 0), *p* = 1.00	9.0 (n = 16)
High blood pressure, % ^3^	41.7 (n = 40)	30.8 (n = 16), *p* = 0.19	29.2 (n = 7), *p* = 0.26	50,0 (n = 3), *p* = 0.69	37.1 (n = 66)
ACR, mg/g	9 (3.5, 36)	45 (14.5, 105) *p* < 0.001	132.5 (72.5, 190) *p* < 0.001	704.5 (596, 797) *p* < 0.001	27.0 (7.0, 88.5)
U-albumin, mg/L	6 (2, 21.5)	53.5 (21, 90) *p* < 0.001	148.5 (77.5, 218.5) *p* < 0.001	762.5 (651, 830) *p* < 0.001	23.5 (5.5, 71.5)
eGFR, mL/min/1.73 m^2^	77 (63.5, 95)	71.5 (53, 100.5) *p* = 0.38	65.5 (35.5, 93.5) *p* = 0.087	55.5 (29, 67) *p* = 0.033	73.5 (54, 94)
P-CRP, mg/L	7.75 (1.75, 40.5)	39.5 (8.85, 126) *p* < 0.001	77.5 (6.75, 179.5) *p* = 0.003	71 (36, 118) *p* = 0.013	20.5 (3.5, 75.0)
WBC, ×10^9^/L	8.045 (6.42, 10.85)	8.775 (7.145, 11.5) *p* = 0.13	9.135 (7.42, 12.15) *p* = 0.20	12.25 (5.2, 16) *p* = 0.31	8.4 (6.7, 11.5)

Values are presented as percentages and absolute numbers in parentheses or median values with their respective interquartile ranges in parentheses, together with *p*-values calculated by *t*-test by comparing the 1+, 2+, and 3+ groups to the 0 or +/− group. CKD, chronic kidney disease; eGFR, estimated glomerular filtration rate; ACEI, Angiotensin-converting-enzyme inhibitor; ARB, angiotensin receptor blocker; NSAID, non-steroidal anti-inflammatory drug; CRP, C-reactive protein. ^1^ Internal medicine patients are defined as patients not admitted as cardiological patients. ^2^ Fever was defined as having a body temperature above 38 °C. ^3^ High blood pressure was defined by a systolic BP > 140 mmHg measured closest to the urinalysis.

**Table 2 diagnostics-12-00457-t002:** Cross-tabling ACR categories with dipstick sub-groups of the study population (n = 178).

ACR-Categories	Dipstick Subgroups				
	0 or +/−	1+	2+	3+	Total
ACR: <30 mg/g	69	21	2	0	92
ACR: 30–299 mg/g	24	30	20	1	75
ACR > 300 mg/g	3	1	2	5	11
Total	96	52	24	6	178

ACR: albumin-creatinine ratio.

**Table 3 diagnostics-12-00457-t003:** Cross-tabling ACR categories with dipstick sub-groups of the follow-up population (n = 35).

ACR-Categories	Dipstick Sub-Groubs				
	0 or +/−	1+	2+	3+	Total
ACR: <30 mg/g	15	9	1	0	25
ACR: 30–299 mg/g	1	3	3	0	7
ACR > 300 mg/g	0	0	1	2	3
Total	16	12	5	2	35

ACR: albumin-creatinine ratio.

**Table 4 diagnostics-12-00457-t004:** Odds ratios for having severe albuminuria (ACR > 300 mg/mL) with a positive dipstick for protein compared with a negative dipstick.

Dipstick for Protein	Odds Ratio	*p*-Value	Confidence Intervals
1+	0.61	0.670	(0.06, 6.00)
2+	2.82	0.272	(0.44, 17.90)
3+	155.0	<0.001	(13.57, 1770.29)

**Table 5 diagnostics-12-00457-t005:** Comparing ACR at admission and at follow-up using ACR > 30 mg/g as positive for albuminuria.

	Follow-Up			Total
Admission		Negative (ACR < 30 mg/g)	Positive (ACR > 30 mg/g)	
	Negative (ACR < 30 mg/g)	17	0	17
	Positive (ACR > 30 mg/g)	8	10	18
total		25	10	35

## Data Availability

Data can be accessed by contacting the corresponding author, on the premise of approval by various Danish national bodies in order to guarantee research integrity.

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
