# Peer review of "Urinary Dipstick Is Not Reliable as a Screening Tool for Albuminuria in the Emergency Department—A Prospective Cohort Study"

_diagnostics, 2022, doi:10.3390/diagnostics12020457_

Round 1

Reviewer 1 Report

Christian B Nielsen et al conducted the observational study to investigate the association urinary dipstick method and UACR in the setting of emergency department. The result of this study is that urinary dipsticks are not 26 reliable to screen for albuminuria in the ED setting.

I think that the result of this study is interesting and meaningful for nephrologist. It is important to detect the risk of kidney failure in the setting of ED. However to evaluate kidney function appropriately and expeditiously is difficult especially in ED department.

This study is well designed and the manuscript is also written from abstract to conclusion.

I suggest one point that it is more meaningful to add the information of urine occult blood. Urine occult blood is widely used for assessing the UTI or urinary stone.

Reviewer 2 Report

  1. AIIA”

Did you mean ARB (angiotensin receptor bloker)?

  1. “During admission, a single mid-stream urinary sample was obtained from all patients both for dipstick and analysis of ACR.”

Was it a first morning urine or just a random spot?

  1. The number of patients especially in the follow-up was rather small.
  2. Dividing patients into smaller groups according to dipstick measurements made the number of patients even smaller therefore the strength of presented statistics is questionable. 

Round 2

Reviewer 1 Report

The revised manuscript is well written and meet the reviewers requirements.  

Reviewer 2 Report

I have no further remarks.